# Tubular Mitochondrial Dysfunction, Oxidative Stress, and Progression of Chronic Kidney Disease

**DOI:** 10.3390/antiox11071356

**Published:** 2022-07-12

**Authors:** Miguel Fontecha-Barriuso, Ana M. Lopez-Diaz, Juan Guerrero-Mauvecin, Veronica Miguel, Adrian M. Ramos, Maria D. Sanchez-Niño, Marta Ruiz-Ortega, Alberto Ortiz, Ana B. Sanz

**Affiliations:** 1Laboratorio de Nefrología Experimental, Instituto de Investigación Sanitaria-Fundacion Jimenez Diaz, Universidad Autonoma de Madrid, 28049 Madrid, Spain; miguel.fontecha@quironsalud.es (M.F.-B.); anam.lopezd@quironsalud.es (A.M.L.-D.); juang.mauvecin@quironsalud.es (J.G.-M.); amramos@fjd.es (A.M.R.); mdsanchez@fjd.es (M.D.S.-N.); mruizo@fjd.es (M.R.-O.); 2Redes de Investigación Cooperativa Orientadas a Resultados en Salud (RICORS) 2040, 28049 Madrid, Spain; 3Institute of Experimental Medicine and Systems Biology, RWTH Aachen University Hospital, 52062 Aachen, Germany; vmiguh00@gmail.com; 4Department of Pharmacology, Universidad Autonoma de Madrid, 28049 Madrid, Spain; 5Department of Medicine, Universidad Autonoma de Madrid, 28049 Madrid, Spain; 6Instituto Reina Sofia en Investigación en Nefrología, 28049 Madrid, Spain

**Keywords:** acute kidney injury, chronic kidney disease, mitochondria, oxidative stress, PGC-1α

## Abstract

Acute kidney injury (AKI) and chronic kidney disease (CKD) are interconnected conditions, and CKD is projected to become the fifth leading global cause of death by 2040. New therapeutic approaches are needed. Mitochondrial dysfunction and oxidative stress have emerged as drivers of kidney injury in acute and chronic settings, promoting the AKI-to-CKD transition. In this work, we review the role of mitochondrial dysfunction and oxidative stress in AKI and CKD progression and discuss novel therapeutic approaches. Specifically, evidence for mitochondrial dysfunction in diverse models of AKI (nephrotoxicity, cytokine storm, and ischemia-reperfusion injury) and CKD (diabetic kidney disease, glomerulopathies) is discussed; the clinical implications of novel information on the key role of mitochondria-related transcriptional regulators peroxisome proliferator-activated receptor gamma coactivator 1-alpha, transcription factor EB (PGC-1α, TFEB), and carnitine palmitoyl-transferase 1A (CPT1A) in kidney disease are addressed; the current status of the clinical development of therapeutic approaches targeting mitochondria are updated; and barriers to the clinical development of mitochondria-targeted interventions are discussed, including the lack of clinical diagnostic tests that allow us to categorize the baseline renal mitochondrial dysfunction/mitochondrial oxidative stress and to monitor its response to therapeutic intervention. Finally, key milestones for further research are proposed.

## 1. AKI and CKD Progression

Chronic kidney disease (CKD) is one of the fastest growing causes of death worldwide, set to become the fifth leading global cause of death by 2040 [1]. Patients with CKD have an increased risk of acute kidney injury (AKI), and AKI may accelerate CKD progression [2]. A better understanding of the molecular mechanisms of kidney injury may allow the development of novel therapeutic strategies. In this regard, recent trials have reported the nephroprotective and cardioprotective impact of sodium-glucose cotransporter-2 (SGLT2) inhibitors, a family of drugs that decrease the transport of molecules in kidney proximal tubules, thus potentially decreasing energy consumption [3]. The kidneys receive 20% of the cardiac output, and proximal tubular cells display a high oxygen consumption rate since they reabsorb the largest portion of the glomerular ultrafiltrate of about 180 L per day. They are also the primary targets in AKI and CKD due to their vulnerability to injury induced by hypoxia, drug-related toxicity, uremic toxins, metabolic disorders, and senescence [4]. In AKI, initial tubular cell death and inflammation is followed by subsequent regeneration [5,6]. The failure of regeneration may lead to the AKI-to-CKD transition [2]. Here, we review the role of mitochondrial dysfunction and oxidative stress in AKI and CKD progression and discuss novel therapeutic approaches.

## 2. Mitochondrial Function in Healthy Kidneys

Mitochondria are intracellular organelles essential to produce adenosine triphosphate (ATP); maintain redox and iron homeostasis; control cell death, inflammation, and intracellular calcium; traffic phospholipids; regulate danger signaling; and synthesize 1,25-(OH)_2_-vitamin D. Therefore, mitochondrial dysfunction can lead to tissue damage and organ failure including AKI and CKD [7,8,9]. Given its critical functions, the homeostasis of mitochondria number and function is key to kidney health. Mitochondria are dynamic and continuously adapt to evolving cell requirements through biogenesis (generation of new mitochondria driven by peroxisome proliferator-activated receptor gamma coactivator 1-alpha (PGC-1α), mitophagy (clearance of damaged mitochondria), fusion, and fission. Disruption of these processes may cause mitochondrial dysfunction.

Kidney tubules are rich in mitochondria because of the high ATP demands required for the absorption of high volumes of ultrafiltrate and solutes. Indeed, the kidney is the second organ with the most mitochondria after the heart; therefore, healthy mitochondria are crucial for normal kidney function [10,11]. Kidney tubules primarily rely on fatty acid β-oxidation (FAO) and mitochondrial oxidative phosphorylation (OXPHOS), likely as an adaptation to generate large amounts of ATP through the electron transport chain (ETC), which is the major source reactive of oxygen species (ROS) in the mitochondria [9] (Figure 1). Electrons released by the ETC react with oxygen to form superoxide anion (O_2_^−^), which is converted to hydrogen peroxide (H_2_O_2_) by superoxidase dismutase (SOD). H_2_O_2_ can be reduced to water by antioxidant enzymes such as catalase and glutathione peroxidases. In this regard, a balance between mitochondrial ROS (mtROS) production and scavenging is critical for a correct mitochondrial function [12]. Increased mtROS production and/or deficient antioxidant defenses may induce mitochondrial dysfunction, leading to cell death, inflammation, and AKI [12]. Tubular cell death induced by different mechanisms, such as cisplatin, calcium oxalate crystals, or ischemia/reperfusion, has been associated to mtROS production and mitochondrial permeability transition pore (MPTP) opening [6,13,14,15,16,17]. The mtROS can activate pro-inflammatory genes, NF-ĸB signaling, and inflammasome proteins [18,19]. Indeed, spontaneous kidney inflammation is observed in PGC-1α-deficient mice [20]. Mitochondrial FAO is the preferred pathway to produce acetyl-CoA as a substrate for the tricarboxylic acid (TCA) cycle in kidney tubules, and FAO dysfunction results in ATP depletion, lipid accumulation, inflammation, and subsequent fibrosis [21,22,23,24]. The mtROS can also affect FAO, as antioxidant strategies restored the expression of proteins involved in FAO in experimental polycystic kidney disease [25]. Antioxidant strategies also improved FAO, lipid accumulation, mtROS-associated lipid peroxidation, and kidney function in murine cisplatin-induced AKI (cisplatin-AKI) [26].

## 3. Mitochondrial Dysfunction and Oxidative Stress in AKI

The key causes of AKI include nephrotoxicity, cytokine storm, and ischemia-reperfusion injury (IRI) [27,28]. Treatments aiming to reduce mitochondrial oxidative stress were protective in preclinical AKI triggered by these causes (Table 1).

### 3.1. Nephrotoxic AKI

Drug-induced nephrotoxicity accounts for up to 60% of hospital-acquired AKI [29]. Several preclinical models have been used to address pathogenic mechanisms in nephrotoxic AKI.

Folic acid-induced AKI (FA-AKI) is a classical model of AKI characterized by tubular cell death, interstitial leukocyte infiltration, and subsequent tubular regeneration and CKD transition; it has been reported in humans [30,31]. Mitochondrial alterations occur at early stages and during AKI-to-CKD progression [21,32,33]. Mitochondrial biogenesis appears to be depressed, along with decreased expression of PGC-1α and its transcriptional regulatory activity [33,34]. Furthermore, FA-AKI is more severe in PGC-1α-deficient mice or those exposed to a mitochondrial complex I inhibitor [34,35]. N-acetylcysteine (NAC) is an antioxidant in clinical use frequently prescribed to prevent radiocontrast nephrotoxicity despite the conflicting results of clinical trials. NAC has been reported to prevent mitochondrial dysfunction in preclinical Huntington’s disease and myocardial infarction [36,37]. In the kidney, NAC pre-treatment in murine FA-AKI prevented mitochondrial and kidney function, but delayed NAC administration increased the severity of FA-AKI [21,32,38]. Variability in intervention timing may be one of the factors underlying the inconclusive results from clinical studies.

The use of the antineoplastic agent cisplatin is limited by nephrotoxicity, especially by AKI that may become irreversible (AKI-to-CKD transition). Preclinical studies support a role for mitochondrial dysfunction in cisplatin-AKI [39,40,41]. Cisplatin nephrotoxicity is characterized by mtROS production, reduced mitochondrial membrane potential, mitochondrial swelling, and loss of mitochondrial function [42]. As in FA-AKI, mitochondrial biogenesis is compromised and PGC-1α expression is decreased in murine cisplatin-AKI [43]. Successful strategies to improve mitochondrial function and to limit preclinical cisplatin nephrotoxicity in mice included SOD mimetics or mitochondrial-targeted antioxidants [42,44,45,46].

Contrast-induced AKI (CI-AKI) is a common cause of AKI in hospitalized patients. During CI-AKI, mtROS increases in association with inflammation and cell death [47,48]. Mitophagy was protective in murine CI-AKI as it removes damaged mitochondria, and rapamycin-induced mitophagy reduced cell injury in cultured human tubular cells [47,49]. The antioxidant compound tetramethylpyrazine preserved kidney function and reduced kidney oxidative stress, inflammation, and aberrant mitochondrial dynamics in CI-AKI in rats [48].

### 3.2. Cytokine Storm

Cytokine storm is a life-threatening organ dysfunction resulting from a systemic inflammatory response to bacterial or viral (e.g., SARS-CoV-2) infection and constitutes the leading cause of AKI in the intensive care setting [50,51,52]. The term sepsis is usually restricted to patients with active bacterial infection; preclinical models frequently involve the administration of sterile bacterial lipopolysaccharide (LPS) or bacteriemia resulting from cecal ligation and puncture (CLP). Cytokine storm-induced AKI (cytokine storm-AKI) is characterized by variable levels of tubular cell death, interstitial inflammatory cell infiltration, and mitochondrial swelling and dysfunction [53,54].

Biopsies from patients with sepsis-associated AKI showed signs of oxidative stress and mitochondrial injury as assessed by the upregulation of oxidative stress markers, mitochondrial DNA (mtDNA) damage, and reduced expression of mitochondrial markers [55]. Moreover, urine mtDNA correlated with mitochondrial dysfunction and AKI severity in sepsis patients [56]. Sepsis causes a metabolic reprogramming in immune cells characterized by reduced mitochondrial OXPHOS and ATP production and increased aerobic glycolysis. This shift promotes a pro-inflammatory phenotype necessary to frame an inflammatory response, but its persistence may promote an exacerbated inflammation, boosting tissue injury [57]. Although there is no clear evidence that tubular cells undergo this metabolic change during sepsis, some data suggest that it may occur [57]. In murine CLP, a change in the metabolite milieu was consistent with reduced flux through the TCA cycle and increased glycolysis in the kidney [58]. Moreover, in murine LPS-induced AKI (LPS-AKI), genes encoding OXPHOS elements were downregulated [59]. Mitochondrial dysfunction in cytokine storm-AKI has been associated with renal failure, which can lead to compensatory quality control mechanisms, as observed in biopsies from sepsis-associated AKI patients, in which genes involved in antioxidant defense, such as SIRT1, in mitochondrial biogenesis, as mitochondrial transcription factor A (TFAM) and PGC-1α and in mitophagy as PTEN-induced kinase 1 (PINK1) and Parkin RBR E3 ubiquitin-protein ligase (PARKIN), were downregulated [55]. This may be interpreted as a scenario of limited generation of new mitochondria and limited removal of damaged mitochondria.

Different strategies against mtROS have been tested in preclinical cytokine storm-AKI. Mitochondria-target ceria nanoparticles have ROS-scavenging activity, improved kidney function, and reduced kidney inflammation and kidney mtROS in murine LPS-AKI [60]. Moreover, both MitoQ, a coenzyme Q10 (CoQ10) analogue, and MitoTEMPO, a SOD mimetic, protected from murine cytokine storm-AKI induced by LPS or CLP, respectively [61,62].

### 3.3. Ischemia-Reperfusion Injury (IRI)

Kidney IRI is a frequent cause of AKI (IRI-AKI) after major surgery and following kidney transplantation. Hypoxia promotes a switch from aerobic to anaerobic metabolism that decreases ATP levels. This is accompanied by a rapid depolarization of mitochondrial membranes and increased intracellular and mitochondrial Ca^2+^ [63,64]. During hypoxia, only a few cells die [64]; however, during reperfusion, multiple mechanisms increase the severity of injury, including oxidative stress resulting from oxygen availability, depletion of antioxidants, and the Ca^2+^-dependent activation of calpains as pH is normalized. Increased mtROS and mitochondrial calcium content lead to MPTP opening and the activation of different pathways of cell death [16]. In addition, mtROS also promote renal inflammation and NLRP3 activation, as in other AKI models [65,66]. The mtROS directly decrease the expression of the mitochondrial transcriptional factor TFAM in tubular cells undergoing hypoxia/reoxygenation and in murine IRI-AKI; in turn, decreased TFAM levels mediate mtROS-induced mitochondrial dysfunction and inflammation in tubular cells, although the effect of TFAM downregulation over cell death was not analyzed [15]. However, it is unknown whether mtROS directly promote kidney inflammation or whether inflammation is mediated by damage-associated molecular patterns (DAMPs) released by dying tubular cells [67].

Pre-treatment with elamipretide (also called Szeto–Schiller 31 (SS-31), MTP-131, and Bendavia) reduced tubular injury and favored kidney regeneration in IRI-AKI in rat, mice, and pigs [68,69,70]. Additionally, CoQ10 nanoparticles and the plastoquinone analogue SkQR1 reduced oxidative stress and tissue injury in IRI-AKI in rats, and MitoQ favored the cold preservation of porcine kidneys [71,72,73]. Taraxasterol, a natural product with antioxidant properties that reduce mtROS in hypoxia/reperfusion-stimulated tubular cells, was also protective in murine IRI-AKI [74]. Administration of isolated healthy mitochondria protected from IRI-AKI in rats and in swine, likely through the uptake and incorporation into cells [75].

Mitochondrial fragmentation, mitophagy, and biogenesis have also been targeted therapeutically. Dynamin-related protein 1 (Drp1) promotes mitochondrial fragmentation, generating mtROS and mitochondrial DAMPs. A specific inhibitor of Drp1, mitochondrial division inhibitor-1 (mdivi-1), attenuated tubular injury and tubular cell death in tubular cells and in murine IRI-AKI [76,77]. Mitophagy protects from kidney IRI by removing damaged mitochondria, and mitophagy activation with an AMP-activated protein kinase (AMPK) activator protected from murine IRI-AKI [78,79,80]. Mitochondrial biogenesis is also reduced in IRI-AKI, and PGC-1α overexpression protected from murine IRI-AKI [20,81]. Moreover, the long-acting beta2-adrenergic agonist formoterol stimulated mitochondrial biogenesis and improved kidney function after murine IRI-AKI [82].

**Table 1 antioxidants-11-01356-t001:** Drugs targeting mitochondrial ROS that have shown beneficial effect in experimental AKI.

Model	Type of Drug	Drug	Ref.
**FA-AKI**	Antioxidant	NAC pre-treatment	[21,32]
**Cisplatin**	CoQ10 analogue	MitoQ	[45]
	SOD mimetics	TEMPOL	[44]
		GC4419	[42]
		Mito-CP	[45]
		MitoTEMPO	[46]
**CI-AKI**	Antioxidant	Tetramethylpyrazine	[48]
**s-AKI**	CoQ10 analogue	MitoQ	[61]
	SOD mimetics	MitoTEMPO	[62]
	Antioxidant	Mitochondria-targeted ceria nanoparticles	[60]
	CoQ10 analogue	SkQR1	[71]
		Mitochondria-targeted TPP CoQ10 nanoparticles	[72]
	SOD mimetics	MitoTEMPO	[15]
	SS-peptide	Elamipretide *	[68,69]
	Isolated healthy mitochondria		[75]
**Reperfusion in experimental atherosclerotic renal artery** **stenosis in pigs**	SS-peptide	Elamipretide *	[70]
**Cold preservation of porcine** **kidneys**	CoQ10 analogue	MitoQ	[73]

Abbreviations: FA-AKI: folic acid-AKI; CI-AKI: contrast induce-AKI; s-AKI: sepsis-induced AKI; IRI-AKI: ischemia-reperfusion injury AKI; NAC: N-acetylcysteine; SOD: superoxide dismutase; CoQ10: coenzyme Q10. * Elamipretide is also called SS-31, MTP-131, and Bendavia.

## 4. Mitochondrial Dysfunction and Oxidative Stress in CKD

There is evidence of mitochondrial dysfunction and related oxidative stress in CKD, as suggested by data in two of the most common causes of CKD, diabetic kidney disease (DKD) and glomerulonephritis.

### 4.1. Diabetic Kidney Disease

DKD is associated with mitochondrial dysfunction in kidney cells including endothelial cells [83] and podocytes [84]. Indeed, abnormal mitochondrial bioenergetics and dynamics precede CKD progression in rat type 1 diabetes (T1D) and altered mitochondrial morphology in tubules of patients with DKD support the idea that mitochondrial injury is an early event in DKD [85,86]. Indeed, in diabetic db/db mice, serum and urine metabolomics revealed differential concentrations of metabolites relevant to energy production by mitochondria, including the TCA cycle, lipid metabolism, glycolysis, and amino acid turnover [87].

The mtROS production in response to chronic hyperglycemia may initiate diverse pathogenic pathways, as was observed in mesangial cells, tubular cells, and in DKD kidneys in rodents [88,89]. Indeed, elamipretide or CoQ10 reduced kidney injury in experimental type 1 (streptozotocin) and type 2 (db/db mice) diabetes (Table 2) [90,91,92]. Post-translational modifications of mitochondrial proteins may also alter mitochondria function and biogenesis [93,94]. Increased nitrotyrosine staining in the kidneys of streptozotocin-induced T1D in ApoE^-/-^ mice was associated with reduced manganese SOD activity [95]. In Madin–Darby canine kidney tubular cells, high glucose culture conditions increased the phosphorylation and oxidation of mitochondrial proteins [96].

A deficiency in the apoptosis-inducing factor (AIF) could contribute to mitochondrial dysfunction and DKD in streptozotocin-induced T1D. In patients with diabetic nephropathy, tubular AIF was decreased and correlated with declining AMP-activated protein kinase (GFR), while AIF overexpression in primary proximal tubule epithelial cells restored OXPHOS capacity under high glucose conditions [97].

The mtDNA released by damaged cells has been proposed as a biomarker of mitochondrial dysfunction. Circulating mtDNA levels were higher in patients with DKD than in healthy subjects or diabetic patients without DKD [98]. These findings were not confirmed by another study that found lower circulating mtDNA in DKD patients than in diabetic patients without DKD [99]. The reason for the discrepancy is unclear and may depend on patient characteristics or comorbidities. However, even when mtDNA was found decreased in plasma, it was increased in urine in DKD patients and in mice with streptozotocin-induced T1D [99]. The mtDNA may play a pathogenic role, as mtDNA infusion induced kidney injury and inflammation in healthy mice.

Altogether, these results suggest that mitochondrial injury may contribute to DKD and mtDNA may be a marker of mitochondrial disfunction in DKD and have a pro-inflammatory role; however, clinical evidence is incomplete. Indeed, the molecular mechanisms of the kidney and cardioprotective effect of SGLT2 inhibitors is not yet fully understood, but there is evidence that they may improve mitochondrial function [100].

### 4.2. Glomerular Disease

In addition to tubular epithelial cells, another epithelial cell type, podocytes, may be marred by mitochondrial dysfunction. Podocyte injury is a key feature of proteinuric glomerular disease, which may also progress to tubulointerstitial injury through proteinuria-induced tubular cell injury. Podocyte injury usually causes a morphological pattern of focal segmental glomerulosclerosis (FSGS) [101]. In humans, several mutations in the nuclear or mitochondrial DNA involving COQ2, COQ6, and PDSS2 and MTTL1, respectively, lead to glomerular disease phenotypes compatible with FSGS and nephrotic syndrome. Moreover, the genetic knockdown of specific mitochondrial genes in animals, such as Pdss2, Tsc1, Mtorc1, Rock1, and Atg5, resulted in proteinuria, glomerulosclerosis, and foot process effacement [102]. The spectrum of mitochondrial functions affected by these genetic defects or manipulations includes mitochondrial tRNA, biosynthesis of CoQ10, protein synthesis control, mitochondrial fission, and mitophagy.

Dysfunction of the energy metabolism and mitochondria may underly several glomerular diseases through deficits in ATP supply, ionic disbalance, unwanted or excessive apoptosis, oxidative stress, and inflammation. Unlike tubular cells, whose energy requirements almost fully rely on OXPHOS, energy generation may shift to glycolysis in injured podocytes during DKD and FSGS [103]. Additionally, a non-glycolytic pathway was uncovered in podocytes under hyperglycemic conditions [104,105]. Under physiological conditions, a deficiency in PGC-1α, Drp1, or TFAM negatively impacted OXPHOS progression in podocytes. However, any defective mitochondrial biogenesis, fission, or mtDNA transcription caused by the deficiency in these factors did not result in mitochondria-associated pathological phenotypes in mice. Thus, podocyte health is maintained when dysfunctional mitochondria lead to mitochondrial respiration deficits. This may be explained by the fact that anaerobic glycolysis was the primary metabolic energy source in podocytes [106]. Nevertheless, a thorough assessment of the impact of mitochondria-associated genetic defects on acquired glomerular injury in vivo is still lacking. Both elamipretide and the SOD mimetic MitoTEMPO were beneficial in high-fat diet-induced glomerulopathy in mice; however, there is limited information regarding their effects on other forms of glomerular injury [107,108] (Table 2). In any case, podocyte injury and albuminuria cause injury of proximal tubular cells, as albuminuria indicates that the capacity of proximal tubular cells to reabsorb filtered albumin has been exceeded. Excessive albumin reabsorption leads to tubular cell injury, inflammatory responses, and loss of the capacity to synthesize Klotho, an antiaging and kidney-protective protein of tubular cell origin [109]. In this regard, *Pgc1α*-KO mice display spontaneous kidney inflammation as evidence of subclinical CKD, while specific proximal tubular TFAM deficiency in mice favors kidney fibrosis through tuna leakage and activation of Sting-dependent innate immune inflammation [20,110].

**Table 2 antioxidants-11-01356-t002:** Drugs targeting mitochondrial ROS that have shown beneficial effects in experimental CKD.

Model	Type of Drug	Drug	Ref.
**STZ-induced diabetic nephropathy**	SS-peptide	Elamipretide *	[90]
**Type 2 diabetes** **(db/db mice)**	SS-peptide	Elamipretide *	[91]
	CoQ10 analogue	CoQ10	[92]
**High-fat diet-induced** **glomerulopathy**	SS-peptide	Elamipretide *	[107]
	SOD mimetics	MitoTEMPO	[108]

Abbreviations: STZ: streptozotocin; CoQ10: coenzyme Q10. * Elamipretide is also called SS-31, MTP-131, and Bendavia.

## 5. Novel Mitochondria-Related Therapeutic Targets in Kidney Disease

In recent years, accumulated evidence suggests a key role of mitochondria-related transcriptional regulators peroxisome proliferator-activated receptor gamma coactivator 1-alpha and transcription factor EB (PGC-1α, TFEB, respectively) and of carnitine palmitoyl-transferase 1A (CPT1A) in kidney disease (Figure 2).

### 5.1. PGC-1α

PGC-1α is the master regulator of mitochondrial biogenesis, a process that coordinates the transcriptional machinery leading to increased mitochondrial mass, allowing tissue adaptation to high energy demands [20,111]. Thus, PGC-1α expression is expected to increase in metabolically demanding tissues such as the kidney [112]. However, PGC-1α levels fall dramatically in kidney diseases, including AKI and CKD [20]. In this regard, PGC-1α downregulation is observed in human AKI of different causes and in experimental AKI models induced by sepsis, IRI, cisplatin, or folic acid, where it is associated with mitochondrial impairment and reduced mitochondrial biogenesis [33,34,59,81,113,114]. Indeed, PGC-1α was the transcription regulator with the most reduced transcriptional activity in FA-AKI [34]. Inflammatory mediators such as TWEAK or TNF-α decreased PGC-1α levels both in vivo and in vitro [33,59]. In fact, neutralizing anti-TWEAK or anti-TNF-α antibodies prevented the downregulation of PGC-1α and its mitochondrial biogenesis-associated target genes in FA-AKI and in sepsis-associated AKI, respectively [33,115]. Additionally, TGF-β1-driven SMAD3 binding overlaps with the active enhancer histone tail modification H3K4me1 of the PGC-1α promoter sequence, blocking the progression of its transcription machinery [116].

By contrast, PGC-1α overexpression increased mitochondrial abundance and protected from AKI induced by sepsis, IRI, and cisplatin in mice [81,114]. PGC-1α overexpression also increases mitochondrial biogenesis in cultured tubular cells [33]. Moreover, PGC-1α enhances the NAD biosynthesis pathway. NAD biosynthesis has emerged as a guardian against the age-related decline in health and mitochondrial function [81] and, potentially, against AKI [117]. PGC-1α also has anti-inflammatory activity since PGC-1α-deficient mice displayed subclinical kidney pro-inflammation and developed more severe FA-AKI [34]. Treatment with the AMPK activator AICAR (5-aminoimidazole-4-carboxamide-1-β-d-ribofuranoside) and the antioxidant ALCAR (acetyl-l-carnitine) increased PGC-1α expression and decreased the severity of cisplatin-AKI by reducing mitochondrial fragmentation, a mechanism that involves Sirt3 activation [43]. Other activators of PGC-1α that also increased mitochondrial biogenesis in preclinical AKI were the Sirt1 inducer SRT1720 in IRI and phosphodiesterase (PDE) inhibitors, which increase cGMP in FA-AKI [118,119].

PGC-1α has been extensively studied in DKD, where PGC-1α activators were nephroprotective in preclinical T1D and type 2 diabetes (T2D) [20]. Thus, pharmacological pyruvate kinase M2 (PKM2) activation and GTPase Rap1b reversed mitochondrial dysfunction in mice with streptozotocin-induced T1D by inducing PGC-1α expression [84,120]. A protective PGC-1α contribution to mitochondrial homeostasis was also observed in glomeruli of T2D BTBR ob/ob mice treated with honokiol, a Sirt3 inducer [121]. In this murine model, podocyte-specific overexpression of Tug1 improved mitochondrial bioenergetics, restoring the expression of PGC-1α and its target genes [122]. PGC-1α overexpression was also protective in tubular cells cultured in high-glucose conditions [123]. In cultured podocytes, Sirt1 agonists such as BT175 and resveratrol protected against high-glucose-mediated mitochondrial injury and reduced oxidative stress through increasing PGC-1α expression [124,125]. However, forced PGC-1α overexpression in podocytes caused collapsing glomerulopathy [126]. CKD is associated to kidney fibrosis; TGF-β1, a key driver of fibrosis, also downregulates PGC-1α expression, leading to lipid accumulation and impaired FAO. Indeed, PGC-1α overexpression in primary murine proximal tubular cells normalized the expression of FAO enzymes after TGF-β1 stimulation [116]. These data support a nephroprotective role of PGC-1α through the improvement of the mitochondrial function in kidney diseases [127]. However, excess PGC-1α expression may not be desirable, at least in some conditions [126].

### 5.2. Transcription Factor EB (TFEB)

TFEB is a master regulator of the autophagy-lysosomal pathway that removes misfolded protein aggregates or damaged organelles. Under baseline conditions, TFEB is retained phosphorylated in the cytosol. Under adverse conditions, TFEB is dephosphorylated and migrates to the nucleus where it promotes the expression of genes of the coordinated lysosomal expression and regulation (CLEAR) network [128]. The TFEB pathway controls mitochondria quality at three levels: mitophagy, mitochondrial biogenesis, and ROS scavenging [129]. TFEB promotes mitophagy by inducing general autophagy upon PINK/PARKIN-induced activation of TFEB, leading to further lysosomal degradation of damaged mitochondria [129,130]. TFEB promotes mitochondrial biogenesis by promoting PGC-1α expression, which, in turn, promotes TFEB expression [129]. Additionally, mtROS promote TFEB activation, and TFEB increases the expression of antioxidant genes such as heme oxygenase 1, SOD2, and thioredoxin 1 [129].

Although TFEB might protect from AKI, direct evidence (i.e., TFEB-deficient tubular cells) is lacking. In PGC-1α-KO mice, kidney TFEB is downregulated, and the number of lysosomes is lower than in WT mice; this was reversed in transgenic mice overexpressing PGC-1α [114]. ZLN005, an activator of PGC-1α expression, alleviates kidney injury in cisplatin-induced AKI; in cultured tubular cells, ZLN005-induced PGC-1α expression has also a protective role partially due to TFEB activation [131]. Moreover, trehalose induces TFEB expression and autophagy and reduces tubular injury and mitochondrial dysfunction induced by cisplatin both in mice and human proximal tubular cells [132]. However, trehalose promoted cell death and inflammation in rat kidney epithelial cells [133]. During murine IRI-AKI, TFEB is activated and upregulates genes of the CLEAR network, promoting autophagy. Indeed, treatment with urolithin A protected from IRI-AKI and promoted TFEB activation [134]. Additionally, the necroptotic protein RIPK3 reduced TFEB expression and autophagy in cytokine storm-AKI, but mitochondrial function was not explored [135].

Regarding CKD, TFEB and its target genes were upregulated in mice with adenine-induced CKD and in patients with IgA nephropathy, but its role was not addressed in interventional studies [133]. However, TFEB was downregulated in human DKD, sub-totally nephrectomized rats, and other forms of kidney fibrosis; this could be mediated by TFEB deacetylation by HDAC6 [136]. Thus, further characterization of the time course of TFEB expression in different models of CKD and the identification of the cell types with upregulated or downregulated TFEB expression are needed to reconcile the current discrepant information.

### 5.3. CPT1A

Experimental and clinical evidence has illustrated the significance of the drastic reduction of FAO coupled with intracellular lipid deposition in proximal tubules in the pathogenesis of both AKI and CKD [137]. CPT1A is the predominant renal isoform of the mitochondrial outer membrane fatty acid shuttling enzyme. It catalyzes the FAO rate-limiting step, i.e., the conversion of long-chain acyl-CoAs into acylcarnitine that is transported into the mitochondrial matrix to undergo FAO [138]. In CKD patients, decreased CPT1A levels are associated to an increased accumulation of short- and middle- chain acylcarnitines, reflecting impaired FAO [139,140]. Both genetic deletion of CPT1A and its pharmacological inhibition with etomoxir uncoupled mitochondrial function and increased the severity of tubular injury and interstitial fibrosis [116,141]. However, the accumulation of fatty acids and lipotoxicity do not appear to be the main drivers of kidney injury associated to impaired FAO. Although the inhibition of CD36-, FATP2-, or KIM-1-mediated fatty acid uptake decreased experimental tubulointerstitial inflammation and fibrosis, increased lipid overload resulting from tubular CD36 overexpression was not sufficient to drive spontaneous renal fibrogenesis [142,143].

Several factors may suppress CPT1A expression in kidney injury. Reduced PGC-1α leads to transcriptional repression of CPT1A and other fatty acid uptake- and oxidation-related genes. The Krüppel-like factor 15 (KLF15) is another transcriptional regulator of CPT1A and Acaa2 that is downregulated in different models of kidney injury, leading to a reduced FAO [144]. Post-transcriptional targeting of CPT1A by miR-33, miR-150, and miR-495 has also been reported in renal epithelial cells under a profibrotic insult [145,146].

Conversely, increasing FAO through genetic or pharmacologic interventions protects tubules from injury. Inducible FAO gain of function through CPT1A overexpression in murine kidney tubules protected from fibrosis in three models of CKD. It also decreased the kidney inflammation and epithelial cell dedifferentiation and preserved structural mitochondrial integrity and ATP generation [139]. There is also more indirect evidence supporting a protective role of FAO restoration: enhanced tubular expression of PGC-1α (a strong inducer of CPT1A) and treatment with fenofibrate (a PPAR-α agonist), C75 (a CPT1A activator and fatty acid synthase blocker), or AICAR (an AMPK activator, which induces CPT1A expression and reduces the levels of its physiological inhibitor, malonyl-CoA) mitigated the chemical or surgical induction of kidney fibrosis and associated functional decline [34,147,148]. Although a protective effect of FAO has been also suggested in AKI, it merits further investigation [59,81,149].

Despite the lack of complete knowledge of the metabolic profile of the renal cell populations involved in fibrogenesis and of whether enhanced FAO is safe in the long term, this strategy has gained momentum as a potential therapeutic approach. The future development of specific, non-toxic CPT1 activators may pave the way for clinical development.

## 6. Therapeutic Modulation of Mitochondrial Dysfunction in Kidney Diseases

Mitochondria-targeted therapies aim to enhance mitochondrial function or to dampen the cellular consequences of mitochondrial dysfunction, including apoptosis, MPTP opening, altered mitochondrial dynamics and mitophagy, or ROS production [20,150,151]. Inhibiting apoptosis-associated mitochondrial pathways was protective in experimental AKI, as exemplified by the genetic deletion of Bak in IRI-AKI. However, other strategies to prevent apoptosis in AKI were not successful; recently, other forms of cell death different than apoptosis, such as ferroptosis and necroptosis, have gained relevance [6,152]. The inhibition of cyclophilin D (CypD) with cyclosporine A or sanglifehrin suppresses MPTP opening and mitochondrial swelling, improving IRI-AKI. However, the therapeutic use of cyclosporine A is limited by its nephrotoxicity [16,153,154,155]. The modulation of mitochondrial dynamics with the mdivi-1, which blocks Drp-1-induced mitochondrial fission, protected from IRI- and cisplatin-AKI [76] but increased the severity of kidney fibrosis [156], evidencing the complexity of interfering with mitochondrial division in injured and regenerating cells [151].

SS peptides, such as elamipretide, are cell-permeable tetrapeptides that bind to cardiolipin on the inner mitochondrial membrane (IMM) and promote ATP synthesis, reduce electron leak and ROS production, and inhibit cardiolipin peroxidation [157]. Elamipretide was renoprotective and restored the mitochondrial structure in preclinical IRI-AKI, experimental DKD, or glomerulopathy (Table 1 and Table 2) [68,69,70,90,91,107]. Additionally, the safety and efficacy of elamipretide in kidney patients was tested in clinical trials (Table 3). A phase 1 trial evaluated the safety of elamipretide in patients with impaired kidney function; however, the results have not been published more than 7 years after its completion (NCT02436447). A phase 2 trial (*n* = 308) evaluated the safety and efficacy of elamipretide in patients with congestive heart failure (NCT02914665), having the impact on kidney function as a secondary outcome. However, no results have been posted. Elamipretide improved kidney function and decreased blood pressure 3 months after angioplasty of the renal artery in a small (*n* = 14) phase 2 trial in patients with atherosclerotic renal artery stenosis [158]. Interestingly, despite elamipretide attenuating renal hypoxia 24 h after contrast imaging and renal artery stent revascularization, it increased peripheral venous levels of G1 cell cycle arrest markers IGFBP-7*TIMP-2 at 24 h after stenting [158]. These biomarkers are associated with AKI, despite the authors’ optimistic interpretation of these findings. In a further small trial (*n* = 24) in heart failure, no temporal change in kidney function was observed associated with elamipretide [159]. The current clinical development program of elamipretide is focused on cardiology, neurology, and ophthalmology, and kidney disease is no longer pursued.

Mitochonic acid (MA-5) also binds to the IMM and increases cellular ATP levels independently of OXPHOS and ETC, promoting survival in fibroblasts isolated from patients with mitochondrial disease. In mice, MA-5 stabilized mitochondrial function, increased ATP content and improved renal function in cisplatin-AKI and IRI-AKI [160]; however, information on clinical trial development is not available on clinicaltrials.gov.

CoQ10 is a component of the ETC. CoQ10 or its analogues are nephroprotective by reducing oxidative stress in diverse forms of kidney injury and are undergoing clinical trials (Table 1, Table 2 and Table 3). CoQ10 protected from DKD in db/db mice and murine IRI-AKI. Indeed, the selective delivery of CoQ10 to mitochondria increased its efficacy compared with free CoQ10 in experimental IRI-AKI [71,72,92]. CoQ10 was safe and well tolerated in hemodialysis patients and decreased oxidative stress in plasma (NCT00908297). Additionally, an ongoing clinical trial (enrolment *n* = 100) is exploring whether pre-treatment with CoQ10 as a dietary supplement before cardiac surgery reduces the incidence of AKI (NCT04445779), while another trial (enrolment *n* = 84) is testing CoQ10 as a dietary supplement to improve renal function after kidney transplantation (NCT04972552). Idebenone, another CoQ10 analog, is already approved in some countries to treat glaucoma and authorized by the EMA to treat Leber’s hereditary optic neuropathy, an inherited disease characterized by progressive loss of sight (https://www.ema.europa.eu/en/medicines/human/EPAR/raxone; accessed on 2 June 2022). Idebenone reduced glomerular injury in murine lupus but has not yet been tested clinically for kidney disease [161]. Among CoQ1 analogs, MitoQ (mitoquinone mesylate) has shown the most promising results. It is currently commercially available as a dietary supplement. In mice, it protected from DKD (db/db mice), from lupus nephritis, and from cisplatin- and LPS-AKI. Furthermore, MitoQ perfusion during cold ischemia before transplantation improved kidney function ex vivo in human and pig kidneys [45,61,162,163,164,165]. An ongoing, small (*n* = 18) clinical trial is exploring the effect of MitoQ as a dietary supplement on exercise capacity in patients with CKD and heart failure (NCT03960073) but will not study its effects on kidney function or markers of kidney injury. Another clinical trial is testing the effect of MitoQ as a nutritional supplement on vascular function and blood pressure reactivity in healthy adults (NCT04334135).

Other SOD mimetic antioxidant strategies tested in preclinical AKI, such as Mito-CP, MitoTEMPO, tempol, and GC441, are not listed in clinicaltrials.gov as being in clinical development (Table 1 and Table 2) [15,45,46,62,108].

**Table 3 antioxidants-11-01356-t003:** Clinical trials related to kidney disease of drugs targeting mitochondria according to clinicaltrials.gov, accessed on 20 May 2022.

Drug (Family of Drugs)	Clinicaltrials.Gov Identifier (Phase)	Title	Disease or Condition	Status
Elamipretide * (SS-peptide)	NCT02436447 (Phase 1)	A Phase 1 Study Investigating the Safety and Pharmacokinetics of Repeat-dose Intravenous Infusion of MTP-131 in Subjects with Impaired Renal Function	Normal and impaired renal function	C
	NCT01755858 (Phase 1, 2) [158]	Effects of Intravenous Bendavia™ on Reperfusion Injury in Patients Undergoing Angioplasty of the Renal Artery (EVOLVE)	Renal artery obstruction, hypertension, renovascular ischemia reperfusion injury	T
	NCT02914665 (Phase 2)	A Phase 2 Study to Evaluate the Cardiac and Renal Effects of Short Term Treatment With Elamipretide in Patients Hospitalized With Congestion Due to Heart Failure	Heart failure	C
CoQ10 (ETC component)	NCT00307996 (Phase 4)	The Effect of CoQ10 Administration on Hemodialysis Patients	Chronic renal failure hemodialysis	C
	NCT00908297 (Phase, not applicable) [166]	Safety and Tolerability of Coenzyme Q10 in Hemodialysis Patients	Cardiovascular disease, ESRD, atherosclerosis oxidative stress	C
	NCT01408680 (Phase, not applicable)	Assessing the Effect of the Dietary Supplement Coenzyme Q10 on Biomarkers of Oxidative Stress, Systemic Inflammation, and Endothelial Function in Hemodialysis Patients	ESRD receiving thrice weekly hemodialysis	C
	NCT03579693 (Phase 2)	Cross-over Randomized Controlled Trial of Coenzyme Q10 or Nicotinamide Riboside in Chronic Kidney Disease	CKD, sarcopenia, frailty	C
	NCT04445779 (Phase, not applicable)	Q10 Preloading Before Cardiac Surgery for Kidney Failure Reduction	AKI	R
	NCT04972552 (Phase, not applicable)	Watermelon/UBIQuinone Study (WUBI-Q Trial)	Kidney transplantation	R
MitoQ (CoQ10 analogue)	NCT02364648 (Phase 4)	Mitochondrial Oxidative Stress and Vascular Health in Chronic Kidney Disease	CKD	U
	NCT03960073 (Phase, not applicable)	Chronic Kidney Disease and Heart Failure with Preserved Ejection Fraction: The Role of Mitochondrial Dysfunction	Chronic renal insufficiency, heart failure with preserved ejection fraction	R
	NCT04334135 (Phase, not applicable)	The Influence of Mitochondrial-Derived Reactive Oxygen Species on Racial Disparities in Neurovascular Function (MAVHS)	Racial disparities, blood pressure, cardiovascular risk factor, renal function	R

Abbreviations: CoQ10: coenzyme Q10; ETC: electronic transporter chain; ESRD: end-stage renal disease; CKD: chronic kidney disease. * Elamipretide is also called SS-31, MTP-131, and Bendavia. Status: C: completed, R: Recruiting, T: terminated.

## 7. Summary and Future Perspectives

In summary, there is increasing preclinical evidence supporting a key role of mitochondria in AKI, the AKI-to-CKD transition, and CKD. However, this has not permeated to the clinic and there are gaps of knowledge regarding the cause-and-effect relationship between mitochondrial injury and kidney injury and the clinical relevance of preclinical studies. The clinical development of some promising agent has stalled (i.e., elamipretide). Others (CoQ10 or its analog MitoQ) are being tested in clinical trials as dietary supplements, which will limit the clinical impact of the results. However, most interventions mainly addressed AKI, while clinical trials exploring kidney protection in CKD are not available and will be difficult to perform. Recent data have identified a key role of mitochondria-related transcription factors (PGC-1α, TFEB) and the fatty acid translocation enzyme CPT1A in kidney disease, unveiling a new set of mitochondria-related therapeutic targets. Another unmet need relates to the lack of clinical diagnostic tests that allow us to categorize the baseline kidney mitochondrial dysfunction/mitochondrial oxidative stress to monitor a patient’s response to therapeutic intervention. The development of such companion diagnostics’ tests should be prioritized to enroll patients most likely to benefit from mitochondria-targeted interventions in clinical trials, to evaluate the pharmacologic effect, to identify responders and non-responders in early phase clinical trials, and to optimize protocols in pivotal trials.

## Figures and Tables

**Figure 1 antioxidants-11-01356-f001:**
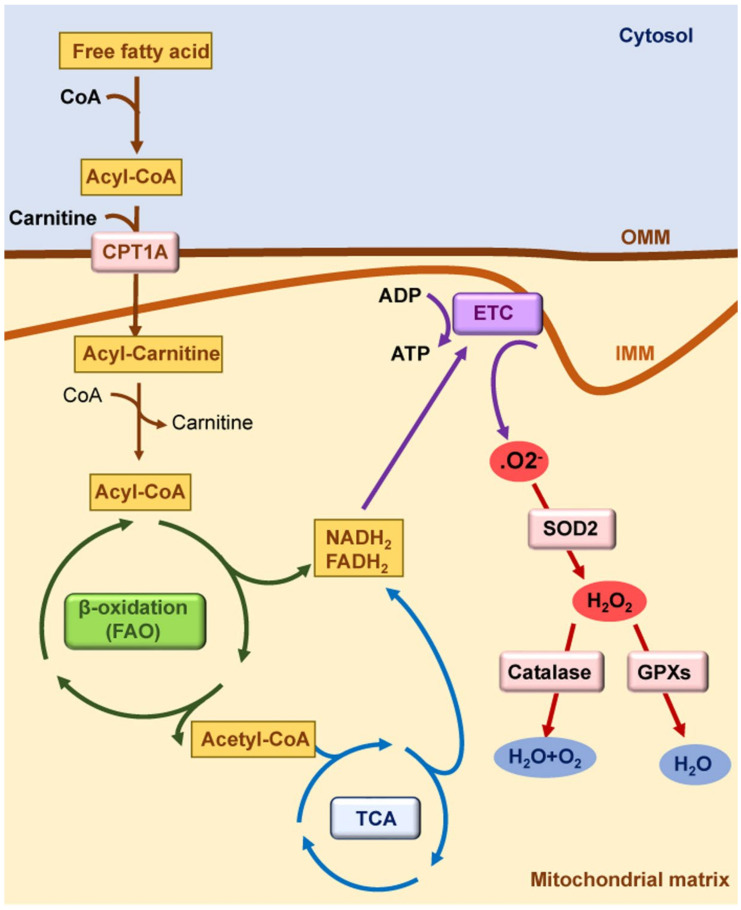
**Mitochondrial function in healthy kidneys.** Mitochondrial fatty acid β-oxidation (FAO) is the preferred pathway to generate ATP in kidney tubules. Fatty acids are converted in acyl-CoA in cytosol; they are conjugated with carnitine in the outer mitochondrial membrane (OMM) by carnitine palmitoyl-transferase 1A (CPT1A) to cross the inner mitochondrial membrane (IMM). In the mitochondrial matrix, acyl-carnitine is reconverted to acyl-CoA, and it enters the tricarboxylic acid (TCA) cycle. Reduced nicotinamide adenine dinucleotide (NADH_2_) and reduced flavin adenine dinucleotide (FADH_2_) generated by FAO and by the TCA cycle deposit their electrons into the electron transport chain (ETC). Electrons released by the ETC react with oxygen to form superoxide anion (O_2_^−^), which is converted to hydrogen peroxide (H_2_O_2_) by superoxidase dismutase 2 (SOD2). H_2_O_2_ can be reduced to water by antioxidant enzymes such as catalase and glutathione peroxidases (GPXs).

**Figure 2 antioxidants-11-01356-f002:**
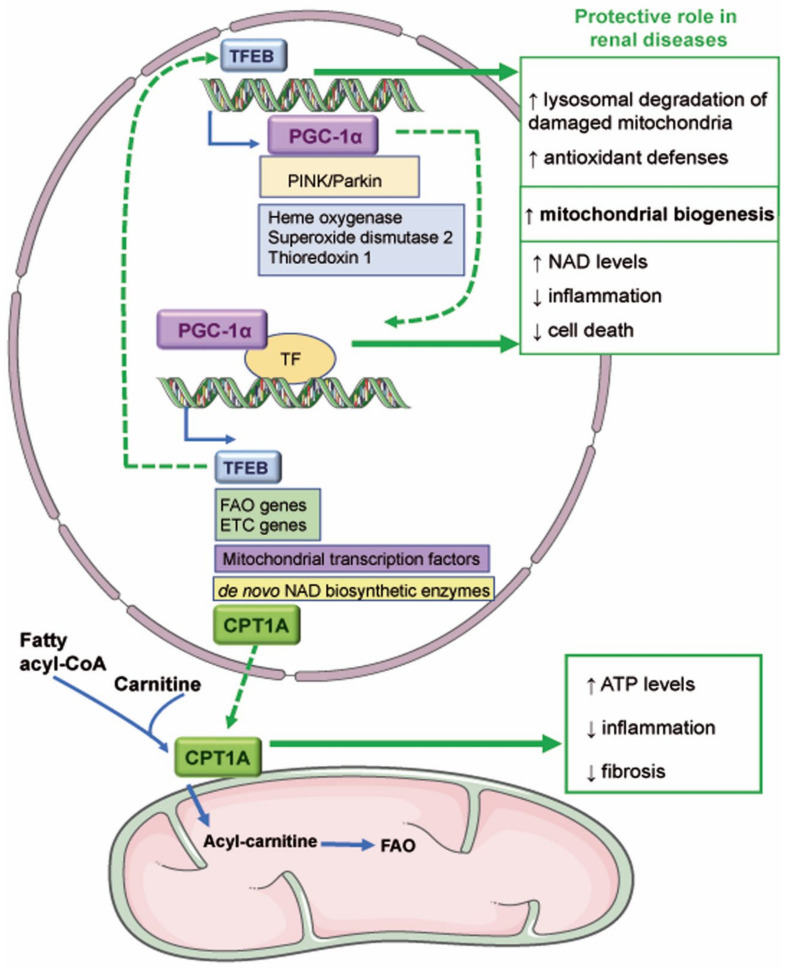
**Novel mitochondria-related therapeutic targets in kidney disease.** The transcriptional activity of TFEB promotes the expression of genes involved in mitochondrial biogenesis such as PGC-1α and in antioxidant defenses. TFEB could be protective in AKI since it favors the degradation of damaged mitochondria and mitochondrial biogenesis. PGC-1α regulates the expression of TFEB, fatty acid β-oxidation (FAO), and electron transport chain (ETC) genes, mitochondrial transcription factors, and de novo NAD biosynthetic enzymes. Various reports have demonstrated that PGC-1α reduces renal inflammation and cell death and favors NAD synthesis and mitochondrial biogenesis. PGC-1α may also mediate the expression of carnitine palmitoyl-transferase 1A (CPT1A), which mediates the transport of fatty acids into the mitochondrial matrix. Overexpression of CPT1A in murine renal tubules protected from preclinical kidney disease by increasing ATP levels and reducing inflammation and fibrosis. ↑ increase; ↓ decrease.

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
