# Peer review of "Tubular Mitochondrial Dysfunction, Oxidative Stress, and Progression of Chronic Kidney Disease"

_antioxidants, 2022, doi:10.3390/antiox11071356_

Round 1

Reviewer 1 Report

In this review article, the authors have provided a summary regarding the role of mitochondrial dysfunction and oxidative stress in different models of AKI and CKD. Moreover, the authors have discussed a few mitochondria-related therapeutic targets and updated the status of clinical development for mitochondria-targeted therapies in kidney disease. The subject of the review is interesting. However, there are some issues need to be addressed:

1.      The abbreviations for all the tables should be placed under the table rather than above the table. For Table 3, the abbreviations for the last column (Status) need to be listed.

2.      The figure legend for Figure 2 needs to be elaborated to cover more information shown in Figure 2.

3.   More careful proofreading is required to avoid the grammar errors, such as:

line 64- “fatty acid” was spelled twice.

Line 84- what’s the number for “32281286”.

Line 147- “along with the expression of PGC-1α and its transcriptional regulatory activity”, should it be “along with decreased expression of PGC-1α and its transcriptional regulatory activity”?

4.      Line 272- “urinary mtDNA was increased in urine from DKD patients and mice with streptozotocin-induced T1D and mtDNA infusion in healthy mice induced kidney injury and inflammation.” The wording here is difficult to follow.

5.      5.2: paragraph 3, regarding the inconsistent expression levels of TEFB in different CKD models, it is suggested to include 1-2 sentences to explain the possible reason for the conflict findings.

6.      References are needed for sentences in Line 287 and Line 521.

Author Response

In this review article, the authors have provided a summary regarding the role of mitochondrial dysfunction and oxidative stress in different models of AKI and CKD. Moreover, the authors have discussed a few mitochondria-related therapeutic targets and updated the status of clinical development for mitochondria-targeted therapies in kidney disease. The subject of the review is interesting. However, there are some issues need to be addressed:

We thank the reviewer for their constructive criticism that has helped to improve the quality of the manuscript. The changes are highlighted in red.

  1. The abbreviations for all the tables should be placed under the table rather than above the table. For Table 3, the abbreviations for the last column (Status) need to be listed.

We thank the reviewer for pointing this out. This has been corrected

  1. The figure legend for Figure 2 needs to be elaborated to cover more information shown in Figure 2.

Following the referee’s advice, we have rewritten the figure legend for Figure 2.

  1. More careful proofreading is required to avoid the grammar errors, such as:

line 64- “fatty acid” was spelled twice. This has been corrected

Line 84- what’s the number for “32281286”. This has been corrected

Line 147- “along with the expression of PGC-1α and its transcriptional regulatory activity”, should it be “along with decreased expression of PGC-1α and its transcriptional regulatory activity”?

This has been corrected

  1. Line 272- “urinary mtDNA was increased in urine from DKD patients and mice with streptozotocin-induced T1D and mtDNA infusion in healthy mice induced kidney injury and inflammation.” The wording here is difficult to follow.

This has been changed for: “urinary mtDNA was increased in DKD patients and mice with streptozotocin-induced T1D while mtDNA infusion in healthy mice induced kidney injury and inflammation.”

  1. 5.2: paragraph 3, regarding the inconsistent expression levels of TEFB in different CKD models, it is suggested to include 1-2 sentences to explain the possible reason for the conflict findings.

Following the referee’s advice, we have included a sentence to comment this contradictory finding.

  1. References are needed for sentences in Line 287 and Line 521.

Following the referee’s advice, we have added the references in these sentences, now are in line 296 and in line 539.

Reviewer 2 Report

I have read the manuscript with interest. The authors reviewed the evidence of abnormal mitochondrial function in the pathophysiology of acute kidney injury and chronic kidney disease and summarized the information about potential drugs targeting mitochondrial dysfunction in kidney diseases. Although mitochondrial dysfunction, oxidative stress, and mitophagy in various kidney diseases was a subject of several recent review articles , the submitted manuscript offers a more cross-sectional perspective and may be worth publishing.

The manuscript is logically arranged and written. However, I think that the authors should be more specific and precisely inform about the sources of evidence (cell culture experiments vs. animal experiments vs. human studies); this concerns especially the sections 3 and 4. In section 4, please clarify which cells you talk about: this is not always easily understandable whether the information relates to tubular or glomerular damage. In conclusions, the gaps of knowledge about the pathophysiological mechanisms related to mitochondrial dysfunction in kidney diseases should be mentioned.

There are numerous English grammar and typing errors that must be corrected. There are some “mental shortcuts” throughout the text (e.g. “PGC-1α enhances the NAD biosynthesis pathway, which has emerged as a guardian against age-related decline in health and mitochondrial function”) that should be corrected. Moreover, please be careful about explaining all the abbreviation on first use, and using them consequently thereafter. A list of abbreviations at the end of the main text would be of value.

Author Response

I have read the manuscript with interest. The authors reviewed the evidence of abnormal mitochondrial function in the pathophysiology of acute kidney injury and chronic kidney disease and summarized the information about potential drugs targeting mitochondrial dysfunction in kidney diseases. Although mitochondrial dysfunction, oxidative stress, and mitophagy in various kidney diseases was a subject of several recent review articles, +the submitted manuscript offers a more cross-sectional perspective and may be worth publishing.

The manuscript is logically arranged and written.

We thank the reviewer for their constructive criticism that has helped to improve the quality of the manuscript. The changes are highlighted in red.

However, I think that the authors should be more specific and precisely inform about the sources of evidence (cell culture experiments vs. animal experiments vs. human studies); these concerns especially the sections 3 and 4.

Following the referee’s advice, we have indicated the source of evidence where this was not indicated.

In section 4, please clarify which cells you talk about: this is not always easily understandable whether the information relates to tubular or glomerular damage.

Following the referee’s advice, we have clarified the cell types that we talk about.

In conclusions, the gaps of knowledge about the pathophysiological mechanisms related to mitochondrial dysfunction in kidney diseases should be mentioned.

We have added this sentence” However, this has not permeated to the clinic and there are gaps of knowledge regarding the cause-and-effect relationship between mitochondrial injury and kidney injury and the clinical relevance of preclinical studies.”

There are numerous English grammar and typing errors that must be corrected. There are some “mental shortcuts” throughout the text (e.g. “PGC-1α enhances the NAD biosynthesis pathway, which has emerged as a guardian against age-related decline in health and mitochondrial function”) that should be corrected.

We have simplified this as follows: Moreover, PGC-1α enhances the NAD biosynthesis pathway. NAD biosynthesis has emerged as a guardian against the age-related decline in health and mitochondrial function [84] and, potentially, against AKI [119].

Moreover, please be careful about explaining all the abbreviation on first use and using them consequently thereafter. A list of abbreviations at the end of the main text would be of value.

An effort has been made to define abbreviation at first use and then use it throughout the rest of the text

Round 2

Reviewer 2 Report

My previous comments have been sufficiently adressed by the authors.